# Integrated decision-control for social robot autonomous navigation considering nonlinear dynamics model

Hui Li[1], Mingyue Luo[1]*, Wanbo Luo[2], Hewei Li[1], Shuofeng Cong[1]

**1** School of Electrical and Electronic Engineering, Changchun University of Technology, Changchun, Jilin, China, **2** Changchun Veterinary Research Institute, Chinese Academy of Agricultural Sciences, Changchun, Jilin, China

☉ These authors contributed equally to this work.

* 1202101007@stu.ccut.edu.cn

**Data availability statement:** All relevant data are within the manuscript and its Supporting information files.

## Abstract

Reinforcement learning (RL) has demonstrated significant potential in social robot autonomous navigation, yet existing research lacks in-depth discussion on the feasibility of navigation strategies. Therefore, this paper proposes an Integrated Decision-Control Framework for Social Robot Autonomous Navigation (IDC-SRAN), which accounts for the nonlinearity of social robot model and ensures the feasibility of decision-control strategy. Initially, inverse reinforcement learning (IRL) is employed to tackle the challenge of designing pedestrian walking reward. Subsequently, the Four-Mecanum-Wheel Robot dynamic model is constructed to develop IDC-SRAN, resolving the issue of dynamics mismatch of RL system. The actions of IDC-SRAN are defined as additional torque, with actual torque and lateral/longitudinal velocities integrated into the state space. The feasibility of the decision-control strategy is ensured by constraining the range of actions. Furthermore, a critical challenge arises from the state delay caused by model transient characteristics, which complicates the articulation of nonlinear relationships between states and actions through IRL-based rewards. To mitigate this, a driving-force-guided reward is proposed. This reward guides the robot to explore more appropriate decision-control strategies by expected direction of driving force, thereby reducing non-optimal behaviors during transient phases. Experimental results demonstrate that IDC-SRAN achieves peak accelerations approximately 8.3% of baseline methods, significantly enhancing the feasibility of decision-control strategies. Simultaneously, the framework enables goal-oriented autonomous navigation through active torque modulation, attaining a task completion rate exceeding 90%. These outcomes further validate the intelligence and robustness of the proposed IDC-SRAN.

## 1 Introduction

In the field of robotics, enhancing the autonomous learning capabilities and natural interaction skills of robots to enable harmonious coexistence with humans in shared physical spaces has become a focal point of research [1]. Autonomous navigation, as a core technology for

**Funding:** This work support from the Department of Science and Technology of Jilin Province (grant number: 20220204090YY). The funders had no role in study design, data collection and analysis, decision to publish, or preparation of the manuscript.

service robots, logistics robots, and other intelligent agents to integrate into human society, is undergoing a paradigm shift from structured environments to dynamic social scenarios, facing the challenge of safe navigation in densely populated environments [2]. Traditional navigation methods based on predefined rules and deterministic optimization, while offering distinct advantages in the interpretability of path generation, often struggle to effectively handle the random behaviors of pedestrian movements [3,4]. The decision freeze represents a fundamental limitation of such approaches in dynamic environments [5,6]. To break through this bottleneck, scholars have proposed an innovative direction of integrating reinforcement learning with traditional planning [7,8]. For example, Pérez et al. [9] by using global path planning to provide waypoint guidance and combining reinforcement learning to handle local pedestrian interactions, the integration of dynamic speed adjustment mechanisms and composite reward functions has successfully addressed the issue of low navigation efficiency in constrained scenarios.

When robots are deployed to perform complex tasks in social scenarios, Reinforcement Learning (RL) based integrated decision-control systems demonstrate unique advantages [10,11] and become the core module for autonomous navigation [12]. Existing RL based integrated decision-control methods primarily utilize the reward function to characterize the behavioral norms of pedestrian [13–15]. A socially aware model was established by Cheng et al. [16], incorporating a reward function that accounts for subtle social norms, thereby enhancing the effectiveness of robotic decision-making and autonomous navigation in dynamic environments. A POMDP approach was proposed in Ref. [17], leveraging reinforcement learning and a novel reward function to achieve adaptive decision-making for social robots. Additionally, two reward functions were designed to enable crowd-aware robotic autonomous navigation [18], focusing on collision avoidance and goal arrival. However, the characteristic that human social behavior is influenced by multiple factors such as culture and emotion makes it essentially limited to manually define the reward function [19].

Over the past few years, the methodology of acquiring reward functions through Inverse Reinforcement Learning (IRL) has garnered significant attention within the academic community. This technique has been strategically applied to refine the efficacy of RL algorithms [20]. This method has achieved significant results in multiple decision-control tasks, such as autonomous flight control [21] and parking autonomous navigation [22]. In the field of autonomous vehicle navigation, a novel method based on IRL was proposed for modeling human-robot collaborative autonomous navigation decisions [23], involving the learning of parameters for a mixture distribution to match observed characteristics. The maximization of expert trajectory probabilities through Maximum Entropy IRL was utilized to derive a reward function, which was then applied to RL to acquire driving strategies more aligned with human driving patterns [24]. IRL was also employed to model the car-following behavior of human drivers, which was subsequently integrated into personalized adaptive cruise control systems [25]. These studies show that IRL technology can effectively break through the representational limitations of empirical reward functions and provide a more reliable solution for exploring social normative reward functions [26].

The core of current research in social robots lies in learning the walking norms of pedestrians to enhance the robots' adaptability and flexibility in social environments. However, the RL methods often simplify the problem, neglecting the dynamics mismatch between pedestrians and robots [27]. These methods overlook the mechanical properties and employ simplistic kinematic models to describe the motion states of robots, which results in a significant sim2real gap [28]. As proposed by Heuthe et al. in Science Robotics, in the research on micro-robot swarm control, counterfactual reward mechanisms must be combined with intrinsic dynamics modeling to achieve precise torque control [29]. The latest research published in

Acta Astronautica indicates that dynamics simulation based on the MuJoCo physics engine is crucial for ensuring the stability of space robot grasping control [30]. Several surveys on reinforcement learning in robotic applications have also confirmed that accurate reproduction of dynamic models is a necessary condition for solving the sim2real problem [31,32]. However, the incorporation of such models introduces significant challenges in RL system design: including the formulation of empirical reward functions accounting for historical states, and issues pertaining to system convergence [33]. During transient transitional phases, temporal discrepancies between state transitions and control actions may emerge due to state delay, potentially generating erroneous reward signals that compromise convergence stability. Furthermore, the prevalent utilization of absolute action definitions in RL systems exacerbates this issue, as abrupt action variations during exploration amplify model transients. Consequently, developing methodologies to counteract transient characteristics remains a critical research priority for implementing robotic models in autonomous navigation decision-control system. In addition, recent studies have enhanced the feasibility of sim2real transfer through perceptual methods [34,35]. For instance, the bird's-eye view semantic conversion method proposed by Reiher [36], and the RetinaGAN image adaptation technique proposed by Ho [37] have demonstrated significant effectiveness. However, this issue is beyond the scope of the current study and may be considered in our future work.

To ensure the feasibility of the decision-control strategy for RL-based socially robots, an Integrated Decision-Control for Social Robot Autonomous Navigation (IDC-SRAN) framework that considers nonlinear dynamics models has been developed in this paper. The contributions of this paper are as follows:

- Pedestrian norm learning based on IRL. To address the issue that pedestrian walking norms are difficult to define through empirical reward functions, IRL is employed to systematically learn pedestrian walking norms. A goal-oriented expert demonstration framework is established, which integrates IRL with a Webots-based training environment for interactive exploration. This methodology enables the identification of both explicit and implicit rewards associated with pedestrian walking norms, thereby enhancing robotic social awareness in complex societal environments.
- Establishment of an Integrated Decision-Control Framework. By introducing a nonlinear dynamic model for the Four-Mecanum-Wheel Robot (FMWR), this study effectively resolves the dynamic mismatch issue prevalent in existing research. Furthermore, the system's actions are defined as incremental torques for the four wheels, which can be directly applied to actual robotic control units. This approach achieves an integrated decision-control for robot autonomous navigation, while enhancing the feasibility of decision-control strategies.
- Design of Driving-Force-Guided Reward Function. Considering the challenge posed by state delays resulting from transient characteristics of the model in reward function design, this study innovatively integrates features of driving force into the reward mechanism. By rewarding driving forces aligned with correct directions, the framework guides robots to conduct proper exploration during transient model phases. This approach ensures task efficiency and safety while significantly reducing the occurrence of suboptimal behaviors.

The paper is organized as follows. Sect 2 presents the four-mecanum-wheel Robot model. Sect 3 described an integrated decision-control framework that considering account social-aware and nonlinear dynamic models. Finally, Sects 4 and 5 are devoted to simulation and concluding remarks, respectively.

## 2 Four-mecanum-wheel robot model

This study focuses on how to integrate wheeled service robots into human social scenarios to provide services. To enhance the social awareness of service robots, it is necessary to simulate the movement patterns of pedestrians, which are characterized by omni-directionality. Therefore, selecting the FMWR, which is capable of omni-directional movement, as the research object is highly meaningful. The symmetric and asymmetric characteristics of the FMWR enable it to have strong adaptability and fast responsiveness in dynamic environments, making it widely used in social scenarios that require high maneuverability and flexibility. This section establishes the kinematic and dynamic models for a FMWR [38], the structure is shown in Fig 1. Let $\{I\} : \{X_i, Y_i, O_i\}$ and $\{R\} : \{X_r, Y_r, O_r\}$ represent the inertial coordinate system and the body coordinate system, respectively.

### 2.1 Kinematic model

Considering the kinematic model error adversely affect the precision of state transitions in reinforcement learning (RL) systems, thereby affecting controller performance. Consequently, establishing high-fidelity dynamic models constitutes a critical step to mitigating the sim2real gap. A MW consists of a fixed standard wheel and passive rollers fixed on the circumference of the wheel, with the roller axes at a 45-degree angle to the hub axis, enabling

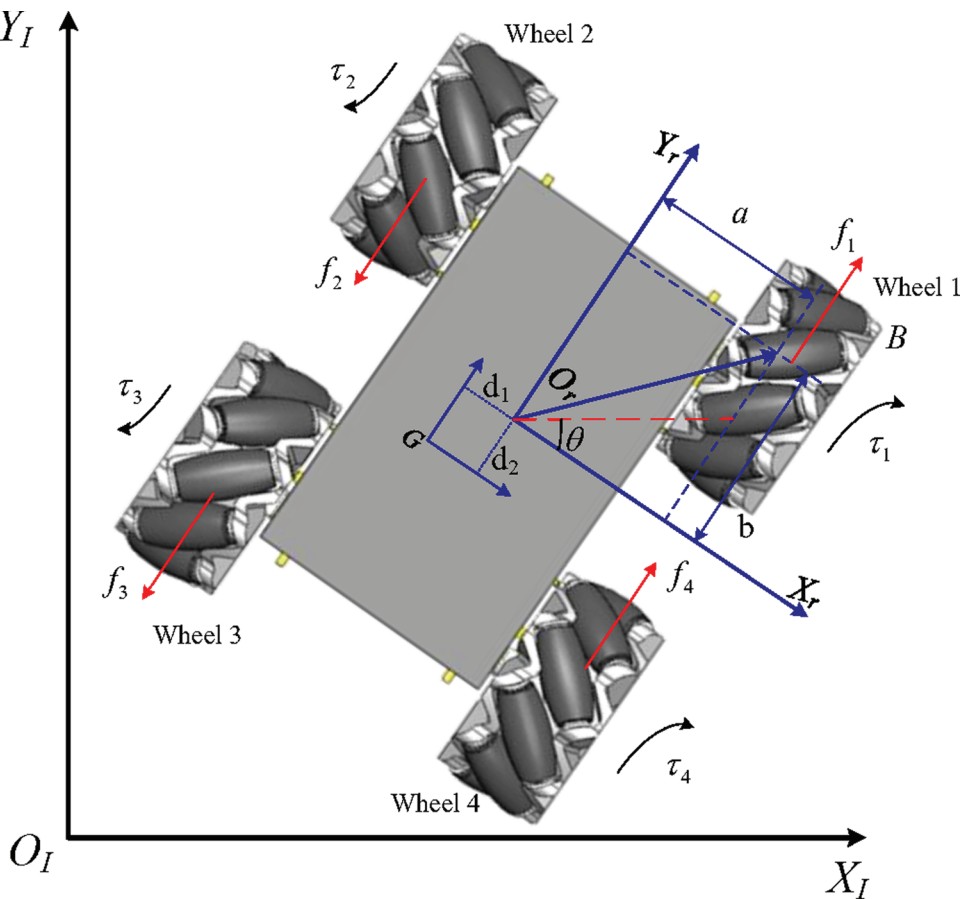

**Fig 1. Schematic of four-mecanum-wheel robot.**

omnidirectional motion [39]. In the inertial frame $\{I\}$, the coordinates are represented as $q = \begin{bmatrix} X_i & Y_i & \theta \end{bmatrix}^T$. Consider a MW with parameters shown in Fig 2. Combining the information from Figs 1 and 2, parameters are defined as follows.

- $m_b$: mass of the platform;
- $m_\omega$: mass of each wheel;
- $I_b$: moment of inertia of the platform about the $Z$ axis;
- $I_\omega$: moment of inertia of each wheel about its main axis;
- $r$: radius of the wheels;
- $a$ and $b$: the distances along the $X_r$ and $Y_r$ directions, respectively;
- $l$: distance from the wheels to the platform's centroid;
- $\alpha$: installation angle of the wheels;
- $\beta$: tilt angle of the wheels;
- $\theta$: rotation angle of the platform;
- $x_G, y_G$: velocity components of the platform's centroid along the $X$ and $Y$ axes;
- $\theta$: angular velocity of the platform;
- $\varphi_i$: rotational speed of the *wheel$_i$*.

Assuming that each MW has an equal radius and installation distance, the inverse kinematics model is:

$$\begin{bmatrix} \dot{\varphi}_1 \\ \dot{\varphi}_2 \\ \dot{\varphi}_3 \\ \dot{\varphi}_4 \end{bmatrix} = -\frac{\sqrt{2}}{r} \begin{bmatrix} \frac{\sqrt{2}}{2} & \frac{\sqrt{2}}{2} & l\sin\left(\frac{\pi}{4}-\alpha\right) \\ \frac{\sqrt{2}}{2} & -\frac{\sqrt{2}}{2} & l\sin\left(\frac{\pi}{4}-\alpha\right) \\ -\frac{\sqrt{2}}{2} & -\frac{\sqrt{2}}{2} & l\sin\left(\frac{\pi}{4}-\alpha\right) \\ -\frac{\sqrt{2}}{2} & \frac{\sqrt{2}}{2} & l\sin\left(\frac{\pi}{4}-\alpha\right) \end{bmatrix} \begin{bmatrix} \cos\theta & \sin\theta & 0 \\ -\sin\theta & \cos\theta & 0 \\ 0 & 0 & 1 \end{bmatrix} \begin{bmatrix} \dot{x}_I \\ \dot{y}_I \\ \dot{\theta} \end{bmatrix} \quad (1)$$

where $\dot{\varphi}_i(i = 1, 2, 3, 4)$ is the rotation speed of the wheel $i$, $r$ is the radius of the main wheel, and $\dot{x}_I, \dot{y}_I, \dot{\theta}$ are the angular velocity in inertial coordinates $\{I\}$.

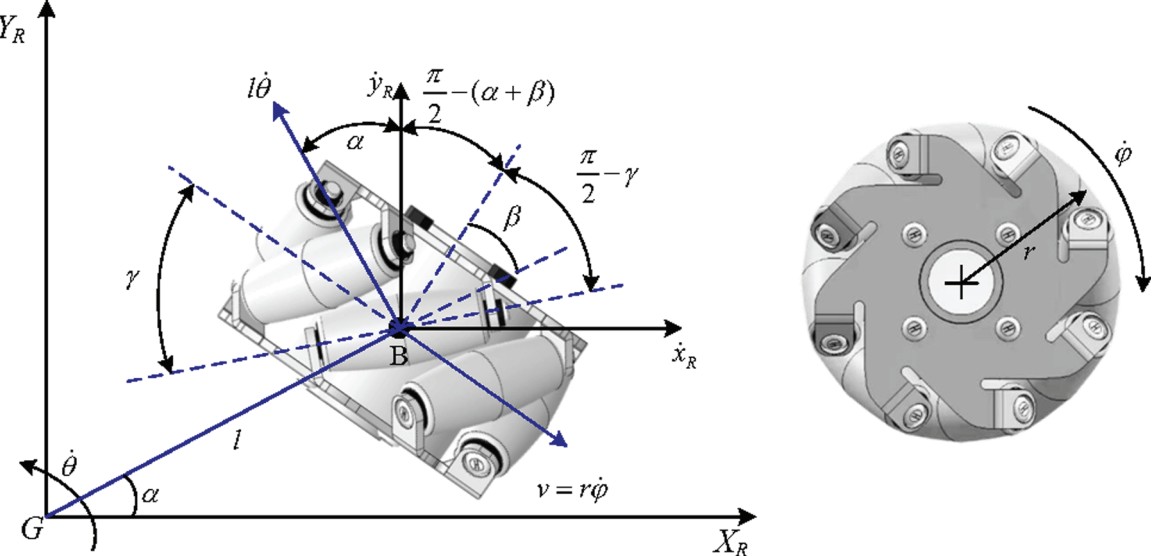

**Fig 2. Parameters of a mecanum wheel.**

## 2.2 Dynamic model

The dynamics model is derived using Lagrange's equation, considering mass eccentricity and friction uncertainty. Consider a four-wheeled Mecanum mobile robot, where $G$ is the geometric center, and $G'$ is the mass center. The total kinetic energy $T$ includes the translational kinetic energy of the platform and the rotational kinetic energy of the wheels:

$$T = \frac{1}{2}m_b(\dot{x}_{G'}^2 + \dot{y}_{G'}^2) + \frac{1}{2}I_{G'}\dot{\theta}^2 + \sum_{i=1}^{4}\frac{1}{2}(m_\omega\varphi_i^2 + I_\omega\varphi_i^2) \tag{2}$$

The Lagrangian $L$ is the difference between the kinetic energy $T$ and the potential energy $V$. Since the robot moves in a plane, the potential energy $V = 0$. Therefore, the Lagrangian is $L = T - V = T$. Using the Lagrange equations:

$$\frac{d}{dt}\left(\frac{\partial L}{\partial \dot{q}_i}\right) - \frac{\partial L}{\partial q_i} = F_i \tag{3}$$

where $q_i$ is the generalized coordinate, and $F_i$ is the generalized force/torque.

The dynamic equations can be expressed in matrix/vector form:

$$M(q)\ddot{q} + C(q,\dot{q})\dot{q} = \frac{1}{r}B^T\tau - B^T S f_0 \tag{4}$$

where $M(q)$ is the mass matrix, $C(q,\dot{q})$ is the Coriolis and centrifugal force matrix, $B$ is the input matrix, $S = diag\{r_1, r_2, r_3, r_4\}$ is the friction matrix, $f_0 = \begin{bmatrix} f_1 & f_2 & f_3 & f_4 \end{bmatrix}^T$ is the friction force vector, $\tau = \begin{bmatrix} \tau_1 & \tau_2 & \tau_3 & \tau_4 \end{bmatrix}^T$ is the input torque vector.

To streamline the model formulation, this paper postulates a uniform mass distribution for the robot, which is a potential limitation that will be the focus in the future research. Therefore, the mass matrix $M(q)$ considering the platform and four Mecanum wheels' mass:

$$M(q) = \begin{bmatrix} m_b + 4m_w & 0 & 0 \\ 0 & m_b + 4m_w & 0 \\ 0 & 0 & m_b + 4m_w \end{bmatrix} \tag{5}$$

This paper ignores the influence of the wheel-surface interactions on Coriolis and centrifugal forces $C(q,\dot{q})$, which are expressed as:

$$C(q,\dot{q}) = \begin{bmatrix} 0 & 0 & m_b\dot{\theta}(d_1\cos\theta - d_2\sin\theta) \\ 0 & 0 & m_b\dot{\theta}(d_1\cos\theta - d_2\sin\theta) \\ 0 & 0 & 0 \end{bmatrix} \tag{6}$$

The input matrix $B$ is:

$$B = \begin{bmatrix} \sin\theta - \cos\theta & -(\cos\theta + \sin\theta) & \sqrt{2}l\sin\left(\frac{\pi}{4} - \alpha\right) \\ -(\cos\theta + \sin\theta) & \cos\theta - \sin\theta & \sqrt{2}l\sin\left(\frac{\pi}{4} - \alpha\right) \\ \cos\theta - \sin\theta & \cos\theta + \sin\theta & \sqrt{2}l\sin\left(\frac{\pi}{4} - \alpha\right) \\ \sin\theta + \cos\theta & \sin\theta - \cos\theta & \sqrt{2}l\sin\left(\frac{\pi}{4} - \alpha\right) \end{bmatrix} \tag{7}$$

Considering that the load may vary, the total mass $m_b$ is divided into a nominal part $m_{b0}$ and an unknown part $\triangle m_b$, i.e., $m_b = m_{b0} + \triangle m_b$. The centroid shift and slipping are also

unknown and can be considered as external disturbances. Therefore, the dynamics model can be expressed as:

$$M_0(q)\ddot{q} = \frac{1}{r}U + D \tag{8}$$

where, $M_0 = diag\{M_1, M_2, M_3\}$, $M_1 = M_2 = m_{b0} + 4\left(\frac{m_w+I}{r^2}\right)$, $M_3 = I_b + 8\left(\frac{m_w+I}{r^2}\right)l^2\sin^2\left(\frac{\pi}{4} - \alpha\right)$, $U = [u_1, u_2, u_3]^T = B^T\tau$, $D = [d_1, d_2, d_3]^T = -B^TSf + \Delta D$, and $\Delta D$ is given by:

$$\Delta D = \begin{bmatrix} \ddot{x}_I & \ddot{\theta}\sin\theta + \dot{\theta}^2\cos\theta & \ddot{\theta}\cos\theta - \dot{\theta}^2\sin\theta \\ \ddot{y}_I & -\ddot{\theta}\cos\theta + \dot{\theta}^2\sin\theta & \ddot{\theta}\sin\theta + \dot{\theta}^2\cos\theta \\ 0 & \ddot{x}_I\sin\theta - \ddot{y}_I\cos\theta & \ddot{x}_I\cos\theta + \ddot{y}_I\sin\theta \end{bmatrix}\begin{bmatrix} \Delta m_b \\ m_b d_1 \\ m_b d_2 \end{bmatrix} \tag{9}$$

This paper focuses on pedestrian movement imitation of robot, where motion control systems demonstrate strong stability in low-velocity regimes (consistent with typical pedestrian locomotion speeds). Consequently, the unknown component of mass is considered to be of negligible magnitude.

## 3 Integrated decision-control framework that considering social-aware and nonlinear dynamic model

In this section, an integrated decision-control framework for social robot autonomous navigation (IDC-SRAN) has been developed, as depicted in Fig 3. The framework constructs Goal-Oriented Expert Demonstration (GOED) and the Training Environment (TE) based on Webots, employing IRL to explore a social reward function that complies with pedestrian walking norms. Subsequently, the system action was modified to an incremental torque, and the actual torque along with the longitudinal-lateral velocity were incorporated into the state space. Then, ensuring the feasibility of decision-control strategy by limiting the increase in torque. Based on the dynamic model, the complete expression for vehicle driving force has been derived, and a driving force-guided reward function, which integrates the dynamic characteristics of the whole vehicle, has been designed. This is intended to mitigate non-optimal behaviours arising from the transient transition processes and to guide the robot to safely and efficiently accomplish its tasks. Ultimately, by integrating A2C algorithm that takes into account the dynamics model of robots, the overall framework was constructed. This framework enables robots to decision-control with socially-aware in complex social environments, thereby ensuring the feasibility of RL strategies.

### 3.1 State and action representation

In social environments shared between humans and robots, the decision-control capabilities of robots are crucial for ensuring their actions are both effective and safe. The feasibility challenge of goal-oriented, socially-aware robot integrated decision-control can be articulated as follows: Given an initial and a target state, the robot must devise decisions that conform to pedestrian walking norms and execute actions conforming to model dynamics, thereby ensuring a smooth transition from the initial state to the target state. Accurately defining the state and action spaces is at the heart of constructing an integrated decision-control framework.

Considering the goal-dependent nature of the robot's behaviour, the state space should encompass information about pedestrians, the target point, and the environmental boundaries. Specifically, the robot's current position is denoted as $(x, y)$, and $(\Delta X_{goal}, \Delta Y_{goal})$ represents the relative distance between the robot and the target. The environmental boundaries are defined by the relative distances between the robot and the stationary obstacles, denoted

**Fig 3. An integrated decision-control framework for social robot autonomous navigation.**

as $(\Delta X_{up}, \Delta X_{down}, \Delta Y_{left}, \Delta Y_{right})$. In the interaction with pedestrians, the primary consideration is given to the information of the four pedestrians who are closest to the robot. The notation $(\Delta X_i, \Delta Y_i)$ represents the relative longitudinal and lateral distances between the $i$-th pedestrian and the robot, respectively. Meanwhile, $(V_{xi}, V_{yi})$ denotes the longitudinal and lateral velocities of the $i$-th pedestrian, respectively. To achieve an integrated decision-control framework, the state and the controller are directly correlated. Defined the actual four-wheel torque of FMWR as $(\tau_1, \tau_2, \tau_3, \tau_4)$, At the same time, it integrates the current lateral-longitudinal velocities $(V_x(t), V_y(t))$ of the robot. The aforementioned states constitute all 30 states of the IDC-SRAN system. This paper assumes all states to be fully observable without unknown disturbances at the perceptual level. This idealized assumption is formulated to facilitate clearer investigation of the model's impact on decision-control feasibility, with refinements to be addressed in subsequent research efforts.

In the definition of the action space, it is considered that the velocity actions learned by RL are difficult to be directly applied to the actual robot controller. Therefore, the system actions are rewritten as the additional torques of the four wheels of the FMWR $(\Delta\tau_1, \Delta\tau_2, \Delta\tau_3, \Delta\tau_4)$.

By restricting the range of the additional torques to $[-1, 1]$, the feasibility of the decision-control strategy is ensured. Given the system's discrete step size of 0.02 seconds, the action range of $\pm 1$ limits the torque rate variation to below 50 $Nm/s$, a specification that is acceptable for the vast majority of robotic control units.

## 3.2 Reward function from expert demonstrations

This section describes the establishment of the Collaborative Interactive Reverse Reinforcement Learning (CIIRL) framework, which utilizes Maximum Entropy Deep Inverse Reinforcement Learning (MEDIRL) combined with GOED and Training Environment (TE) to explore pedestrian motion norms, i.e., to explore the social reward function of pedestrian interactive motion. The goal of robots is to complete tasks in a safe and effective manner. Imitating human behavior patterns helps to improve the acceptability of their behavior.

In the execution of tasks, robots require a clear goal point. However, in real-world scenarios, the destinations of pedestrians are often unknown, which implies that existing expert demonstration data lacks information pertaining to goal-directed behaviour. Therefore, the states of GOED, in addition to their own information, should also include all details of the goal, pedestrians, and static obstacles. This paper employs the ZARA scenarios from the ETH dataset as expert demonstration samples, which provide pedestrian coordinate information at intervals of 0.4 s. In the experimental setup, a pedestrian from the scene is randomly selected as the target agent. Fig 4 shows a frame of its motion process. The current position of the agent is marked with a red circle, while its trajectory and terminal coordinates are represented by the red dashed line and solid dot, respectively. The boundary box outlined by the light blue dashed lines quantifies the spatial distance between the agent and the scene's edges. Based on the nearest distance principle, the four pedestrians most closely interacting with the agent are selected and their current positions are marked with green, yellow, pink, and orange circles, respectively. The trajectories and terminal coordinates of these interacting objects are represented by the dashed lines and solid dots in the corresponding colors. It is worth noting that the pedestrian targets are unknown in real-world scenarios. However, in the limited environment of the TE, defining the terminal coordinates within the scene as the task goal is engineeringly reasonable. This approach is consistent with the logic of autonomous driving systems, which preset waypoints in known topological maps [40]. In this manner, an expert demonstration dataset containing target point information, referred to as the GOED, is constructed. This approach is pivotal for the IRL process, which is employed to learn the underlying reward functions that reflect pedestrian walking norms and to enhance the performance of robots in complex social environments.

Upon the Webots platform, a training environment has been constructed for the purpose of training policy and value networks. The perimeters of the TE are meticulously delineated through a suite of multivariate linear equations. To simulate a high-density social milieu, the TE incorporates a simulation of fifteen dynamic pedestrians. The genesis of these pedestrians within the confines of the environment is subject to randomisation, thus guaranteeing a variety of scenarios and introducing an element of unpredictability. The pedestrians are instigated with initial velocities in both the longitudinal and lateral axes, which are arbitrarily selected within the parameters of $(v_{x_i}, v_{y_i}) \in [-1, 1]$. The simulation time step is set at 0.05 seconds, during which a random increment in pedestrian velocity, both longitudinal and lateral, will be introduced at each step $(\Delta v_{x_i}, \Delta v_{y_i}) \in [-0.02, 0.02]$. Upon interface with the environmental confines, pedestrians are considered to have attained their intended destination. Subsequently, the system imparts novel coordinates and initial velocities to these agents in

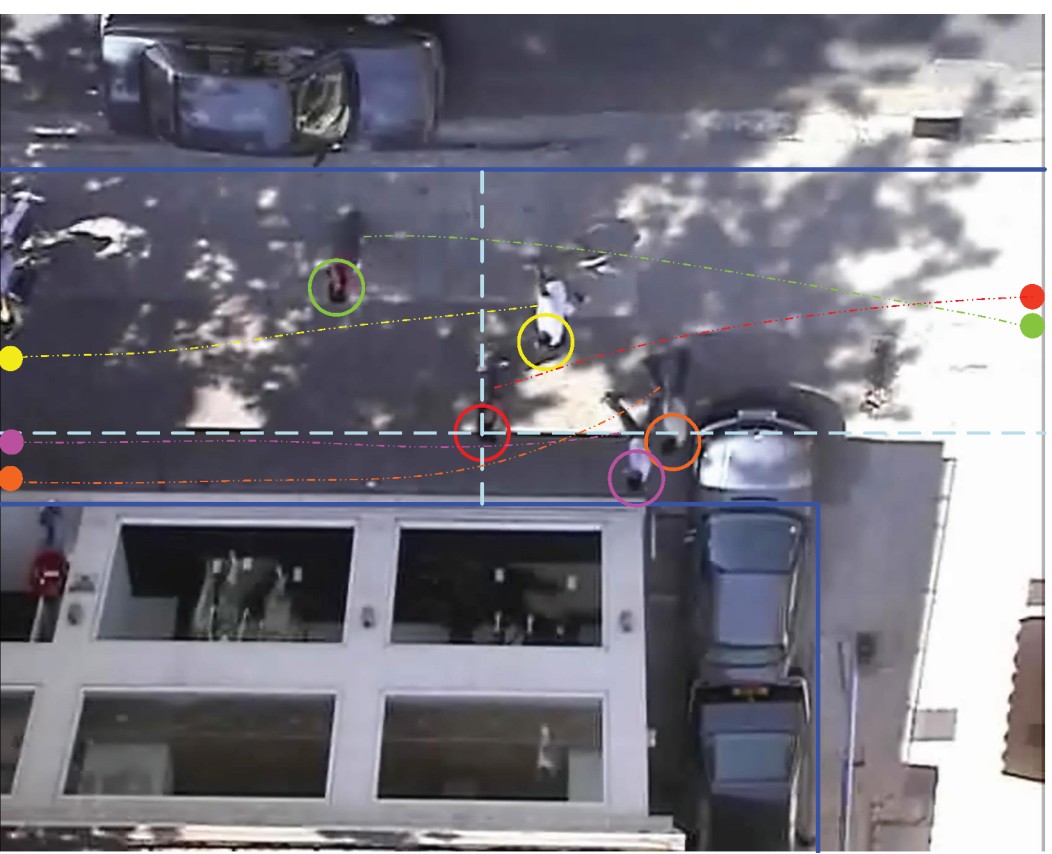

**Fig 4. The ZARA scenarios from the ETH for GOED.**

a stochastic fashion, facilitating their reintroduction into the simulated milieu for the purpose of further engagement in social dynamics. This design introduces an element of randomness to the pedestrians' movement, ensures a degree of continuity in their direction of travel, thereby emulating the natural walking patterns of pedestrians in the real world.

During the process of employing IRL to infer reward functions from expert demonstrations, there arises a potential issue of ambiguity in the interpretation of rewards. To mitigate this ambiguity within the reward function, the principle of maximum entropy is applied to guide the exploration of rewards, thereby circumventing the risk of overfitting to the demonstrated data. The MEDIRL algorithm reformulates the constrained optimisation problem into a maximum entropy model. The ultimate objective is to identify a reward function $R$ for the policy $\pi$ that closely mirrors the performance of goal-oriented expert demonstrations $\pi_E$. The discrepancy between the prior policy and the expert demonstration is articulated through the sum of squared probability errors, denoted as $(\pi_E - \pi)^2$.

$$Loss(\theta) = \frac{\partial \mathcal{L}}{\partial R(\theta; s, a)} = \pi_D(s, a) - \pi(s, a) \tag{10}$$

Therefore, the final gradient of the objective function $\mathcal{L}$ with respect to the parameter $\theta$ is,

$$\frac{\partial \mathcal{L}}{\partial \theta} = (\pi_E - \pi)\frac{\partial R_\theta}{\partial \theta} + \frac{\partial \mathcal{L}_2}{\partial \theta} \tag{11}$$

Upon the completion of each training round, an evaluator interacts with the training environment through the policy $\pi$, which is trained with the current episode's reward function $R_s$. After 480 training episodes, the reward loss associated with intReractions with the GOED has been reduced from 1 to approximately 0.05, as illustrated in Fig 5. Although the reward loss associated with interactions with the TE diminishes over time, achieving convergence to a minimal and tolerable threshold remains elusive.

### 3.3 Driving-force-guided reward function

In the realm of RL-based robotic motion control, the nonlinear characteristics of the dynamic model, especially in the transient transition stage of velocity changes caused by torque, have a significant impact on the dynamic performance of the robot. To effectively neutralize the problems caused by transient characteristics, and to accurately reflect the dynamic nature of robotic movements while reducing misleading reward signals, a driving-force-guided reward function has been meticulously designed. This function encourages the robot to apply forces in alignment with the target direction, thereby ensuring efficient task execution, meanwhile incorporates strategies for safe motion that preemptively avoids collisions.

The driving force of the robot can be expressed as, which can be derived from Equation (4).

$$F = \frac{1}{r}B^T\tau - B^T S f_0 = B^T(\frac{\tau}{r} - S f_0) = B^T F \tag{12}$$

where $F = \begin{bmatrix} F_1 & F_2 & F_3 & F_4 \end{bmatrix}^T = \begin{bmatrix} \frac{\tau_1}{r} - f_1 & \frac{\tau_2}{r} - f_2 & \frac{\tau_3}{r} - f_3 & \frac{\tau_4}{r} - f_4 \end{bmatrix}^T$.

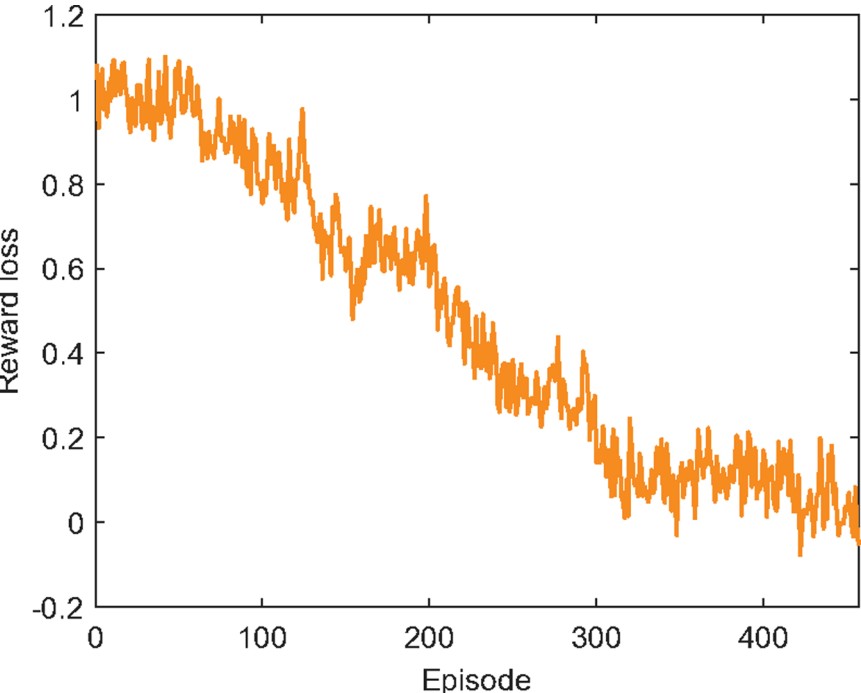

**Fig 5. The reward loss associated with interactions with the GOED.**

According to Newton's second law of motion, it follows that.

$$[M_1\dot{v}_x \quad M_2\dot{v}_y \quad M_3\dot{\theta}]^T = B^T F \tag{13}$$

Hence, it is possible to derive the longitudinal and lateral driving forces of the robot, which are respectively expressed as:

$$F_x = b_{11}F_1 + b_{21}F_2 + b_{31}F_3 + b_{41}F_4 \tag{14}$$

$$F_y = b_{12}F_1 + b_{22}F_2 + b_{32}F_3 + b_{42}F_4 \tag{15}$$

To enable the robot to reach the target point more swiftly, it is desired that the force acting upon the robot be directed towards the target point. Define the angle between the direction of the force and the target direction as $\alpha$.

$$\vec{d} \cdot \vec{F} = \|\vec{d}\| \cdot \|\vec{F}\| \cos\alpha \tag{16}$$

$$\cos\alpha = \frac{(x_g - x)F_x + (y_g - y)F_y}{\sqrt{(x_g - x)^2 + (y_g - y)^2}\sqrt{F_x^2 + F_y^2}} \tag{17}$$

Where the vector representing the force acting on the robot is denoted as $\vec{F} = (F_x, F_y)$ and its magnitude is represented as $\|\vec{F}\| = (F_x^2 + F_y^2)^{\frac{1}{2}}$. The vector representing the direction from the robot's current position $(x, y)$ to the target point $(x_g, y_g)$ is given by $\vec{d} = (x_g - x, y_g - y)$. The magnitude of this vector is $\|\vec{d}\| = ((x_g - x)^2 + (y_g - y)^2)^{\frac{1}{2}}$.

Hence, a reward function based on the direction of the driving force is defined. If the cosine of the angle is positive, it indicates that the direction of the force is aligned with the target direction, and thus the reward is positive. Conversely, if the cosine is negative, it suggests that the force is directed opposite to the target direction, resulting in a negative reward.

$$R_g = \|\vec{F}\| \cos\alpha \tag{18}$$

This implies that the reward is at its maximum when the direction of the force is aligned with the target direction, which corresponds to $\cos(\alpha) = 1$.

During the execution of its task, a robot inevitably interacts with pedestrians in its vicinity and must also consider the avoidance of other obstacles (including static obstacles and dynamic pedestrians). Given that the social behaviour rewards associated with interactions with pedestrians have been explored and determined through the method of MEDIRL, this part does not explicitly differentiate between dynamic pedestrians and static obstacles in the analysis. Let the current position of the robot be denoted as $(x, y)$, and the position of the $i$-th obstacle be represented as $(x_i, y_i)$.

When the direction of the robot's movement aligns with the relative position of an obstacle, the obstacle avoidance mechanism is activated to ensure that the robot can take appropriate evasive actions. If not, the robot will prioritise following its preset trajectory towards the target destination. Specifically, the robot's force-directed local collision avoidance reward function will only be considered within the context of the avoidance strategy when the direction of the robot's movement matches the position of the obstacle, that is, when both $v_x(x - x_i) < 0$ and $v_y(y - y_i) < 0$ are satisfied simultaneously. The collision avoidance reward function

is designed as follows:

$$R_{o-x} = \begin{cases} -|\vec{F}_x|, & v_x(x - x_i) < 0 \ and \ v_x \cdot F_x > 0 \\ |\vec{F}_x|, & v_x(x - x_i) < 0 \ and \ v_x \cdot F_x < 0 \end{cases} \quad (19)$$

$$R_{o-y} = \begin{cases} -|\vec{F}_y|, & v_y(y - y_i) < 0 \ and \ v_y \cdot F_y > 0 \\ |\vec{F}_y|, & v_y(y - y_i) < 0 \ and \ v_y \cdot F_y < 0 \end{cases} \quad (20)$$

Here, $R_{o-x}$ denote the reward function for the robot's lateral avoidance of obstacles. When the robot's longitudinal velocity relative to the pedestrian is in the opposite direction of the relative distance, i.e., $v_x(x - x_i) < 0$, it indicates that the robot is moving towards the pedestrian. In this case, if the velocity component $v_x$ and the force $F_x$ are in the same direction, i.e., $v_x \cdot F_x > 0$, it suggests that the robot's motion towards the pedestrian is accelerating, which may pose a collision risk or cause discomfort to the pedestrian. The system will then impose a penalty. Conversely, if the velocity component $v_x$ and the force $F_x$ are in opposite directions, i.e., $v_x \cdot F_x < 0$, it indicates that the control system is effectively implementing deceleration to avoid the pedestrian, and the system will provide a positive reward. Similarly, a reward function $R_{o-y}$ has been designed for robots to lateral avoid obstacles.

The collision avoidance reward must take into account the relative distance to the obstacles, which can be encapsulated as follows:

$$R_o = \frac{\delta}{\Delta d}(R_{o-x} + R_{o-y}) \quad (21)$$

The relative distance is $\Delta d = \sqrt{(x - x_i)^2 + (y - y_i)^2}$.

Furthermore, discrete rewards such as collision penalties $R_{coll}$ and task completion rewards $R_{com}$ are jointly deployed within the IDC-SRAN framework, which can be expressed as:

$$R_{coll} = \begin{cases} 1, & \Delta d < 0.5m \\ 0.5, & \Delta d < 1.2m \\ 0, & else \end{cases} \quad (22)$$

$$R_{com} = \begin{cases} 1, & \|\vec{d}\| < 0.1m \\ 0, & else \end{cases} \quad (23)$$

In summary, the reward function $R$ of IDC-SRAN can expressed as,

$$R = R_f + c_3 R_{coll} + c_4 R_{com} + c_5 R_s \quad (24)$$

$$R_f = c_1 R_G + c_2 R_o \quad (25)$$

Where $c_i$ are referred to as the reward weighting constants.

## 3.4 Simulation environment

To train the proposed IDC-SRAN framework and validate its autonomous navigation capabilities, this study leveraged the Webots simulation platform to construct a testing and validation environment that replicates the real-world environment of the ETH dataset, as illustrated in Fig 6.

Fig 6(a) depicts the real-world environment of the ETH dataset, and Fig 6(b) presents the testing and validation simulation environment with the Webots simulation platform. The

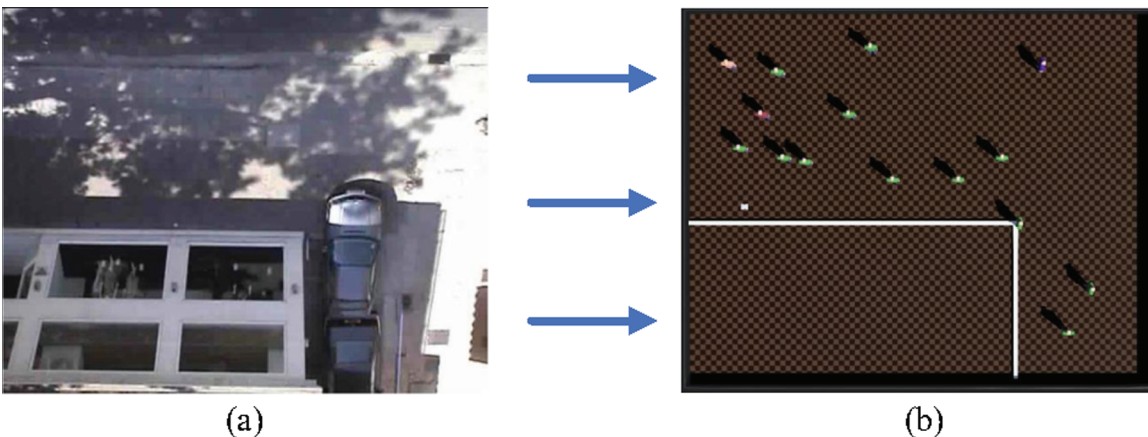

**Fig 6. Simulation environment for testing and validation (Scene 1).** (a) Real-World Environment of the ETH Dataset. (b) Simulation Environment with Webots

white lines represent walls enclosing a rectangular structure in the lower-left area, resembling commercial facilities in real-world environments with pedestrian-impermeable boundaries. It is noteworthy that the simulation environment does not constitute a 1:1 replication of the real-world environment. Specifically, parked vehicles along road edges are treated as pedestrian-impermeable boundaries in the simulation, and the depth of the lower-right corner is deliberately deepened. This design modification reduces corner width while increasing its length, thereby enhancing the operational complexity for robots to pass through the corner. Such modification further facilitate the training and validation of rationality in autonomous navigation decision-control strategies. The simulated environment measures 21.6 meters in width and 17.4 meters in length, with the commercial facilities in the lower-left area measures 15.5 meters in width and 7.7 meters in length, containing a total of 15 pedestrians.

Furthermore, this study is grounded in the assumption of global observability, neglecting the research on the perception module. Consequently, IDC-SRAN remains constrained to simulation environments and faces significant challenges in practical deployment on real-world robotic platforms. Notably, limited scene generalization capability emerges as one of the critical obstacles in practical implementation of autonomous navigation decision-control systems. Therefore, an additional verification environment was developed based on existing conditions, which is shown in Fig 7, precisely aligned with a real-world corridor to validate the generalization performance of IDC-SRAN. The additional verification environment features a rectangular configuration with a length of 12 meters and a width of 8.5 meters, containing a total of 10 pedestrians. The movement of all pedestrians within the environment is governed by the social force model, which continuously calculates their velocity and position at each discrete time step. Motion control in the simulated environment is ultimately executed through the Supervisor node of the Webots platform, enabling precise simulation of pedestrian dynamics.

### 3.5 Advantage actor-critic

Considering the inherent advantages of the Advantage Actor-Critic (A2C) algorithm, which enables parallel synchronous updates and typically achieves higher training efficiency with fewer hyperparameters, and it avoids hyperparameters adjustments associated with Proximal

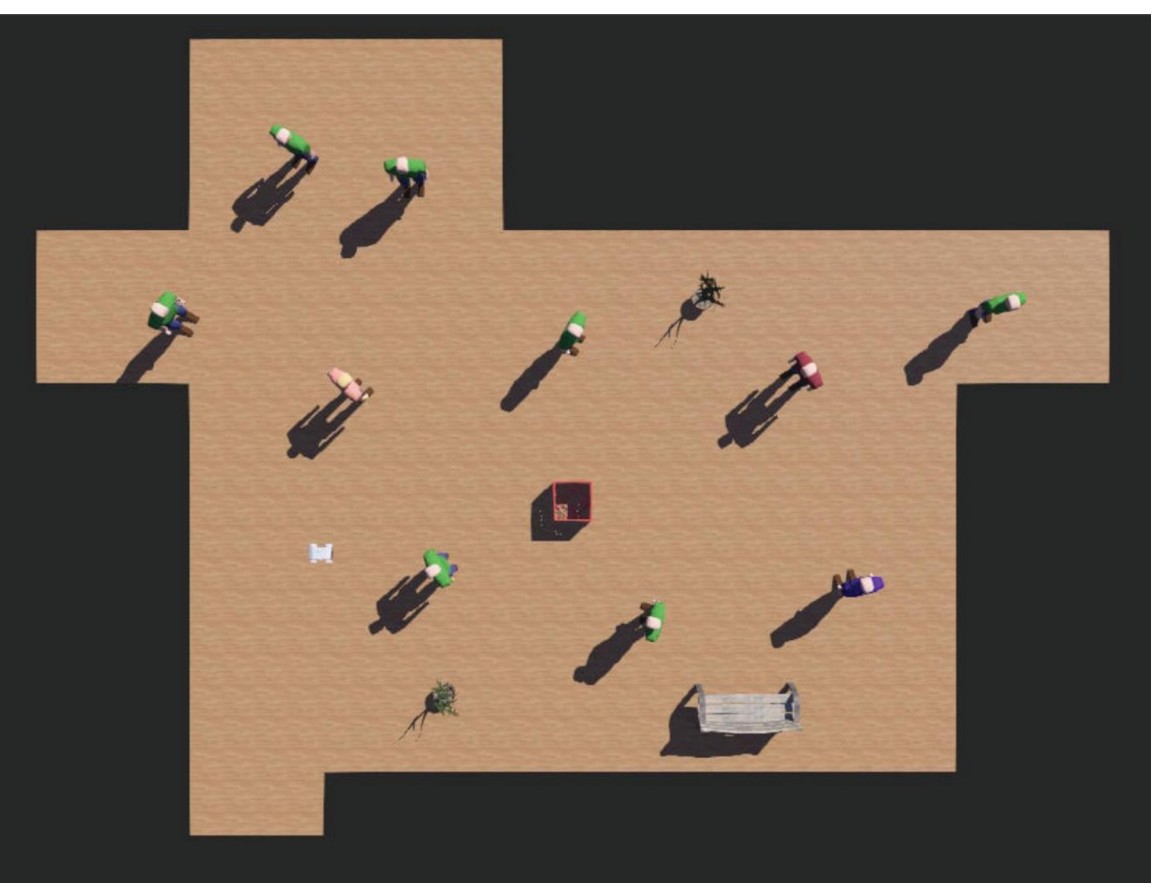

**Fig 7. Additional verification environment for Real-World Corridor (Scene 2).**

Policy Optimization (PPO) clipping ranges or entropy coefficient balancing in Soft Actor-Critic (SAC). To optimize both computational efficiency and hyperparameters adjustments simplicity, the A2C is employed to construct the IDC-SRAN framework. The advantage function of A2C can be mathematically expressed as,

$$A^\pi(s_t, a_t) = Q^\pi(s_t, a_t) - V^\pi(s_t) = R + \gamma V^\pi(s_{t+1}) - V^\pi(s_t) \tag{26}$$

Then, the policy gradient becomes:

$$J(\pi_\theta) = E\left[\sum_{t=0}^{T} A_\phi(s_t, a_t) \nabla_\theta \log \pi_\theta(a_t|s_t)\right] \tag{27}$$

The policy and value networks consist of a three-layer DNN, each with 128 neurons in the hidden layers. Among them, both the policy network and value network feature an input layer dimension of 30, representing the number of states in the IDC-SRAN framework. Their output layer dimensions of 4 and 1 correspond to the action number and the policy evaluation value, respectively. ReLU activation functions are used across all networks, and the optimizer is Adam with a weight decay of $10^{-4}$. The learning rates are $5*10^{-5}$ for the reward network and $10^{-4}$ for A2C, the discount rate is 0.99.

In Fig 8(a) and 8(b), the blue lines distinctly illustrate the training loss under the IDC-SRAN framework, encompassing both policy loss and value loss as pivotal metrics. In parallel, the orange lines depict the training loss of the baseline IDC framework without deploying $R_f$, maintaining identical experimental settings. During training, iterative updates were

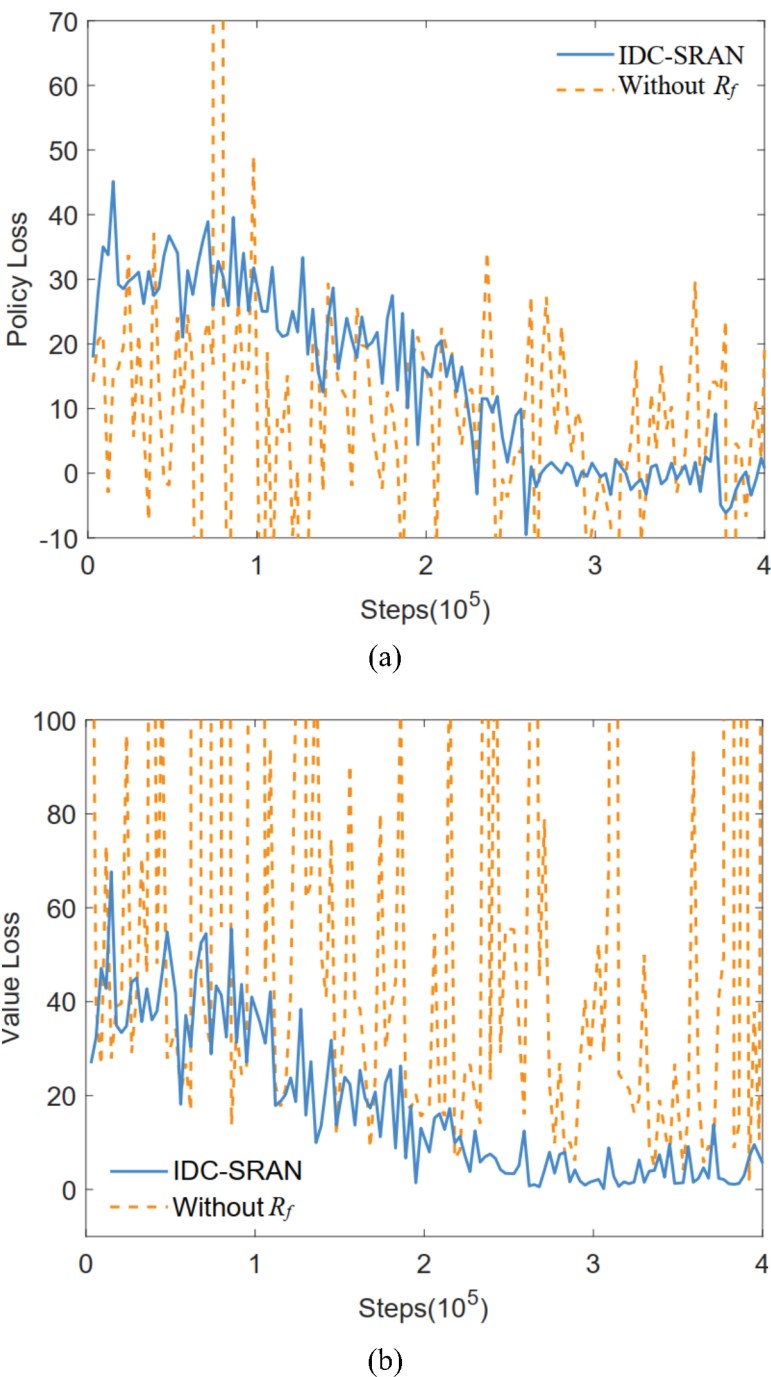

**Fig 8. Comparative analysis of IDC-SRAN framework: policy and value loss with and without $R_f$.**

performed every 3000 steps, with a convergence criterion defined as the average of policy and value losses over ten consecutive iterations being less than 1. Clearly, IDC-SRAN demonstrates significant advantages in convergence speed and stability. Furthermore, to further validate training efficacy, the Total Average Return (TAR) is presented in Fig 9, which is derived from the evaluation phase after every iteration. While the TAR trajectory of the IDC framework without $R_f$ exhibits a gradually increasing trend, it is accompanied by pronounced oscillations. Combined with the loss oscillations illustrated in Fig 8 demonstrate that state delay caused by transient characteristics of robot model hinders the agent's to correctly associate rewards with the actions generating them. By contrast, the introduction of $R_f$ enables rational exploration during model transients, leading to marked improvements in both stability and convergence.

## 4 Experiments

In this section, the trained network is implemented online within the constructed scenario to validate its autonomous navigation capabilities. The focus is on ensuring that the robot's interactions with pedestrians in a human-robot coexistence environment adhere to pedestrian locomotion norms. To ensure the feasibility of the reinforcement learning strategy, it is crucial to ensure the smoothness of the robot's trajectory and the continuity of its velocity changes.

Employing a social reward function as delineated in Sect 3.2, in conjunction with the A2C algorithm presented in Sect 3.4, an autonomous decision-making RL (ADM-RL) framework is constructed. This framework considers robots as a moving mass point while neglecting its dynamics model, defining longitudinal and lateral velocity as the action space. Consequently, during comparative experiments with IDC-SRAN, motor energy consumption differences between the two frameworks were not analyzed in this paper. Fig 10(a) illustrates the motion trajectories of the robot and pedestrians within the environment, with black dots indicating

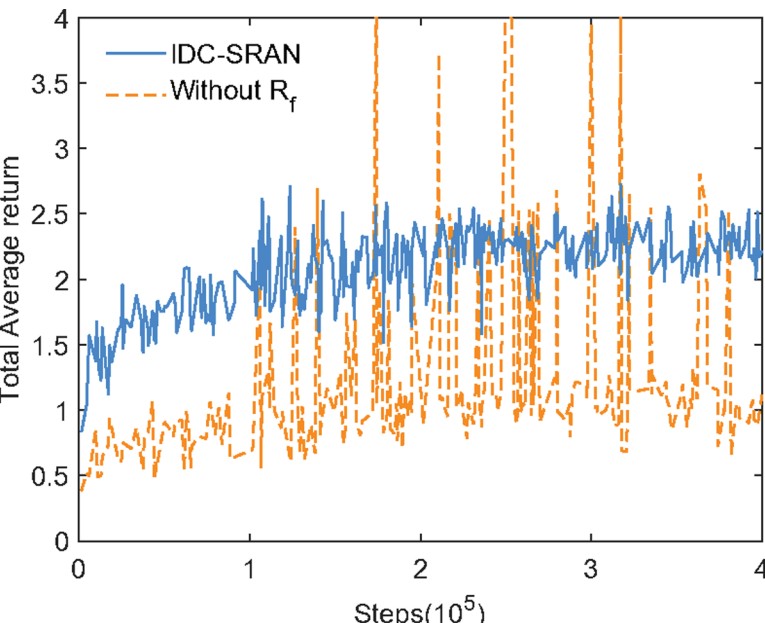

**Fig 9. Comparative analysis of IDC-SRAN framework: total average return.**

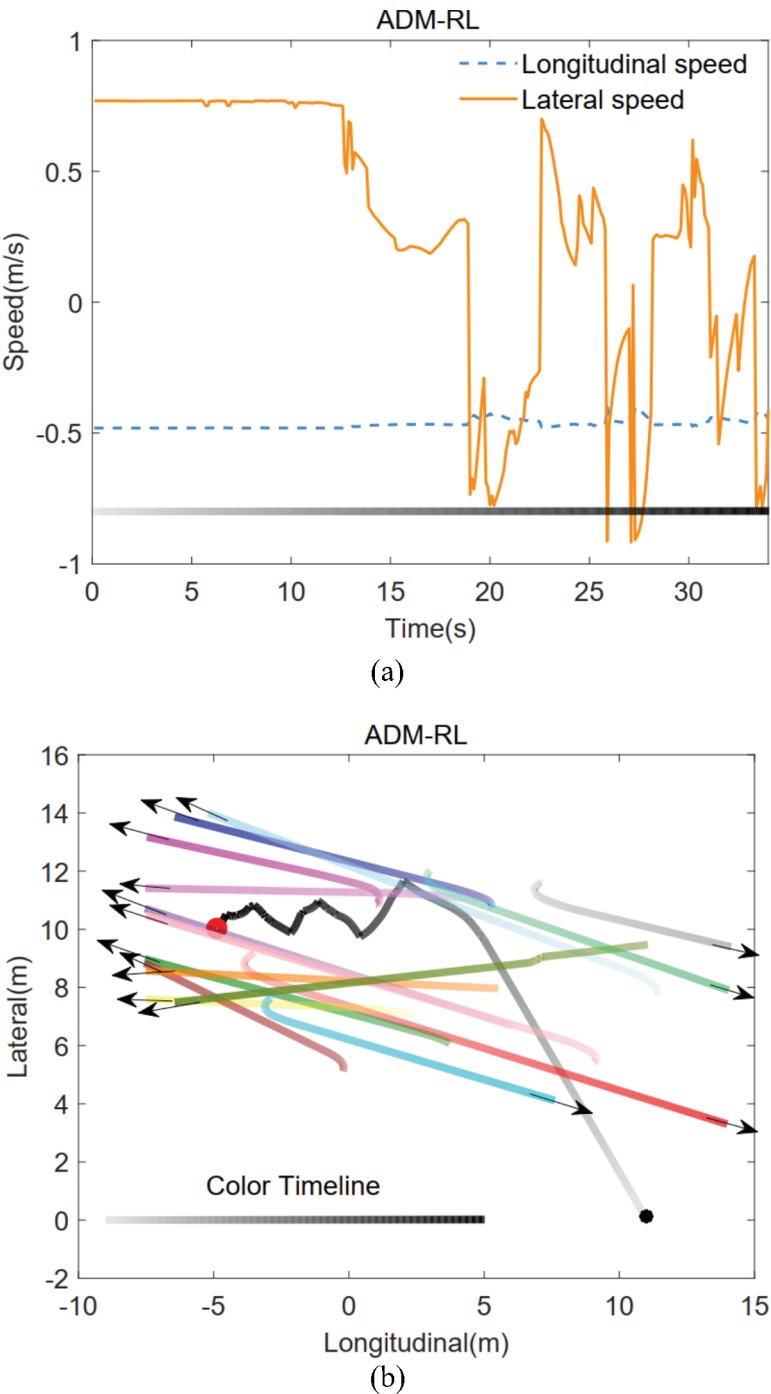

**Fig 10. Without considering robot dynamics in ADM-RL framework in Scene 1: the trajectory of pedestrian and robot, and the robot velocities.**

the starting points and red dots signifying the endpoints. Complementing this, the velocity variation curves in Fig 10(b) reveal that, in the initial phase of the environment devoid of pedestrians, the robot advances towards the target at a constant velocity. Upon entering

densely populated areas, the robot effectively interacts with pedestrians through a continuous decision-making process, ultimately reaching the designated target successfully. Although this method achieves commendable results in simulation, the velocity action still requires domain controllers for tracking in practical deployment, and its step changes introduce substantial challenges for maintaining stable tracking performance in real-world scenarios.

Fig 11(a) and 11(b) illustrate the pedestrian movement trajectories and velocity variation curves within the IDC-SRAN framework. In this framework, actions are defined as four-wheel torque inputs, while the model outputs velocity values that serve as environmental states for system decision-control. The gradient color variations in the figure represent the temporal evolution of the robot's and pedestrians' positions. As demonstrated in Fig 11(b), the black and orange trajectories (corresponding to the robot and pedestrian) approach collision when the color intensity reaches approximately two-thirds of the defined range. Through proactive adjustments to the four-wheel torque, the robot reduces its lateral velocity around 10 seconds, successfully evading the collision. This outcome underscores the intelligence of the IDC-SRAN decision-control framework in adaptive collision avoidance. Notably, IDC-SRAN effectively enhances the smoothness of lateral and longitudinal velocities. However, as a decision-control scheme, it eliminates the need for real-time prediction of dynamic pedestrians' behaviors or optimal path planning. This design results in a compromise in path optimality compared to conventional planning-based tracking algorithms. Nonetheless, this does not imply that the path optimality of the decision-making control approach cannot be improved, which will be further studied in future research.

To quantitatively analyze the impact of IDC-SRAN and ADM-RL on velocity smoothness, acceleration-related data from different algorithms in Scenario 1 are presented in Figs 12 and 13. The acceleration profile of ADM-RL illustrated in Fig 12(a), after entering the crowded area, the robot exhibited drastic velocity fluctuations while maneuvering to avoid pedestrians and proceed toward the goal. Notably, most velocity adjustments were executed within a single discrete time step, resulting in frequent sudden acceleration changes. While these actions demonstrate the intelligence of decision-control, effectively preventing collisions and maintaining correct goal-oriented movement, their practical implementation faces limitations due to actuator dynamic response constraints, thereby compromising the feasibility of realizing such actions in real-world scenarios. Fig 13 the bar chart comparing the mean and maximum absolute acceleration of both IDC-SRAN and ADM-RL. Under identical scenarios, IDC-SRAN demonstrates notably rational acceleration characteristics, with its maximum instantaneous acceleration capped at 1, which is only 8.3% of the peak value by ADM-RL. This substantial reduction in acceleration extremes not only validates the feasibility of IDC-SRAN's decision-control strategy but also enhances pedestrian interaction experience, as smooth acceleration transitions help prevent startling pedestrians. This outcome further corroborates the intelligence of IDC-SRAN in adaptive control and autonomous navigation of robot.

As shown in Fig 14(a), the variation curves of the robot's four-wheel torques under IDC-SRAN framework are presented. To more intuitively demonstrate the impact of the four-wheel torque changes on the robot's dynamic model, the variation curve of the robot's driving force is shown in Fig 14(b). Referring to Fig 7, it can be observed that in the initial stage, the goal of the robot is located in the upper right. The appropriate four-wheel torques provide a positive lateral force and a negative longitudinal force, which gradually increase the robot's lateral and longitudinal velocities in the correct direction. This phenomenon is also demonstrated in Fig 12(b), which shows negative longitudinal acceleration and positive lateral acceleration.

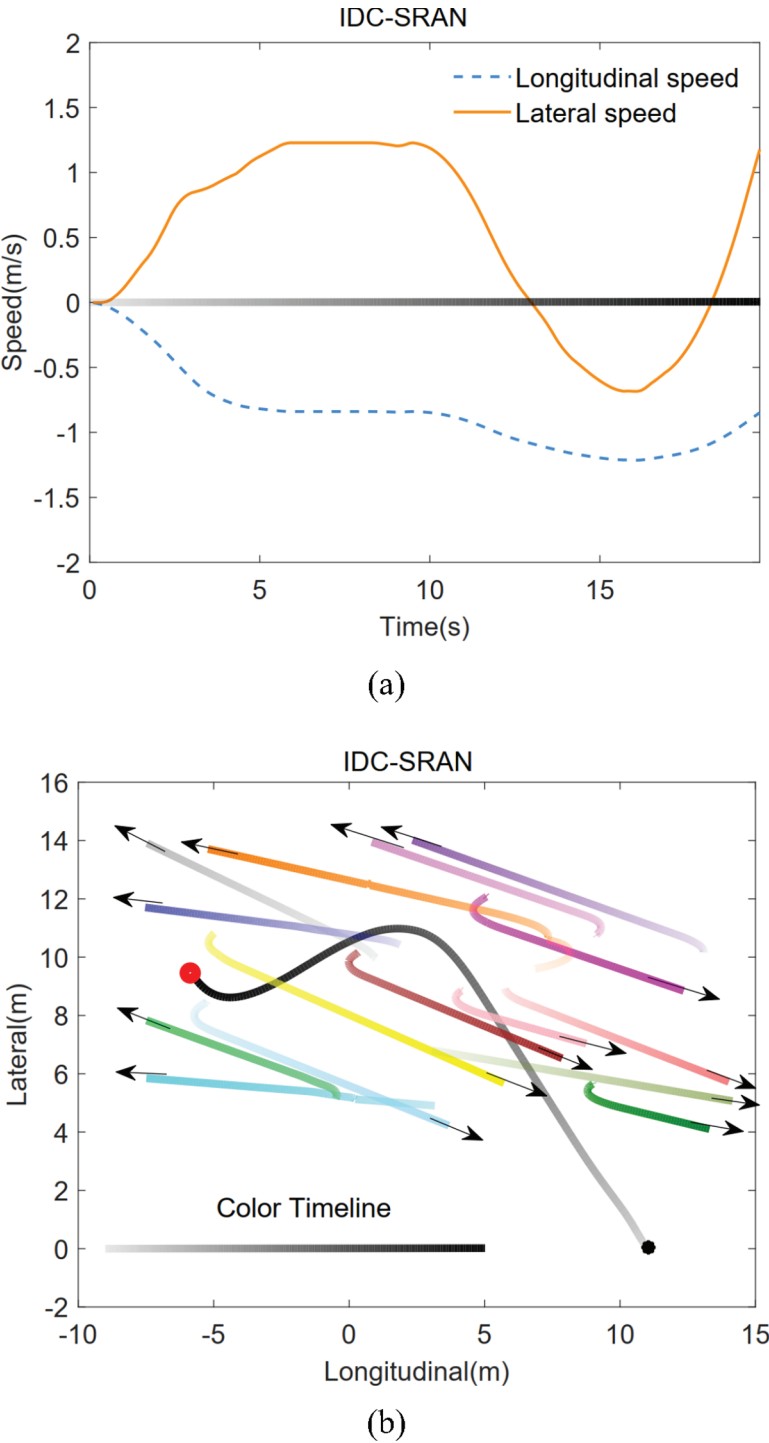

**Fig 11. With considering robot dynamics in IDC-SRAN framework in Scene 1: the trajectory of pedestrian and robot, and the robot velocities.**

About 9 s, the goal is directly to the right of the robot, and the positive lateral velocity continues to move the robot to the upper right. Therefore, the agent provides the robot with a negative lateral force through the four-wheel torques to reduce the velocity and further approach

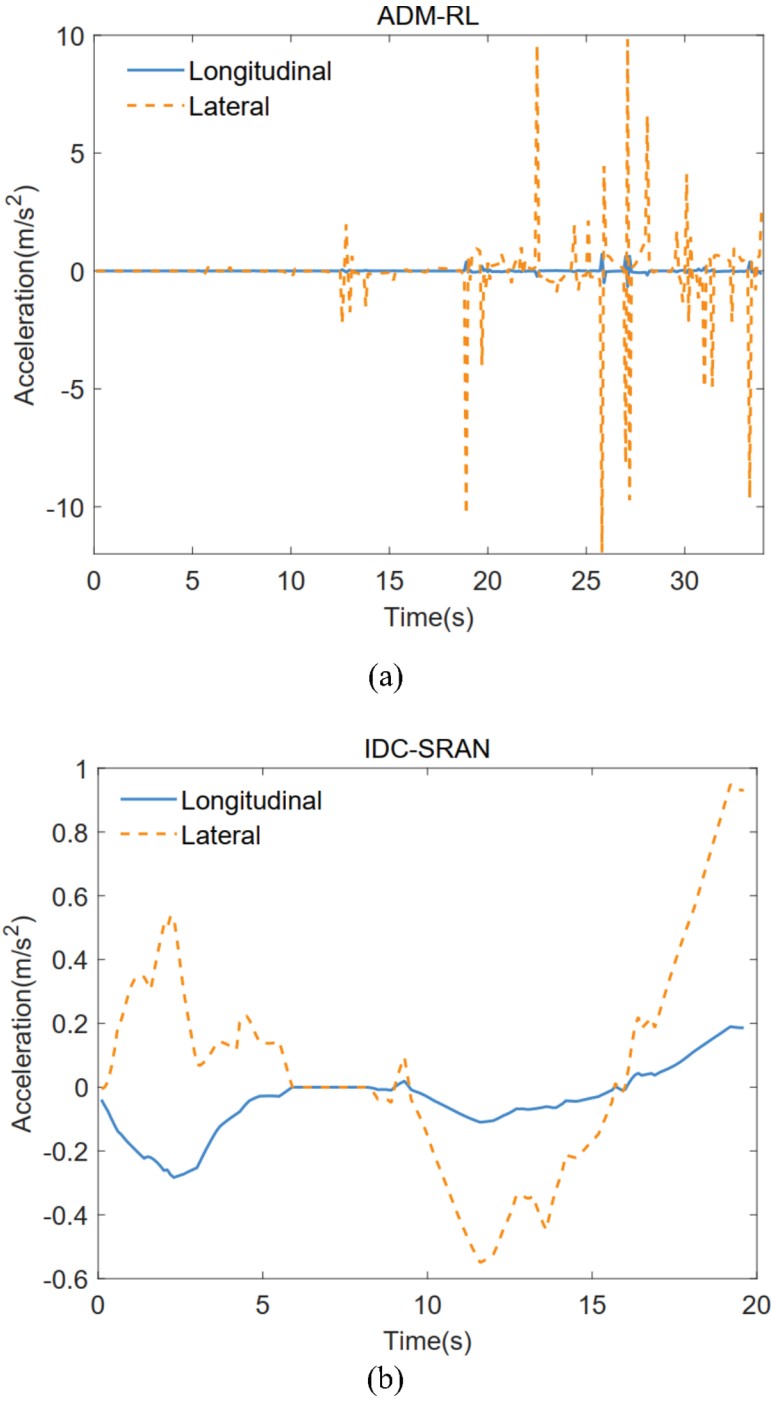

**Fig 12. The accelerations of ADM-RL and IDC-SRAN in Scene 1.**

the goal. The subsequent process follows the same principle. Clearly, this driving method is more in line with the actual characteristics of the motor and has stronger strategic feasibility compared to Fig 10. To maintain modeling simplicity, the motor model was excluded from

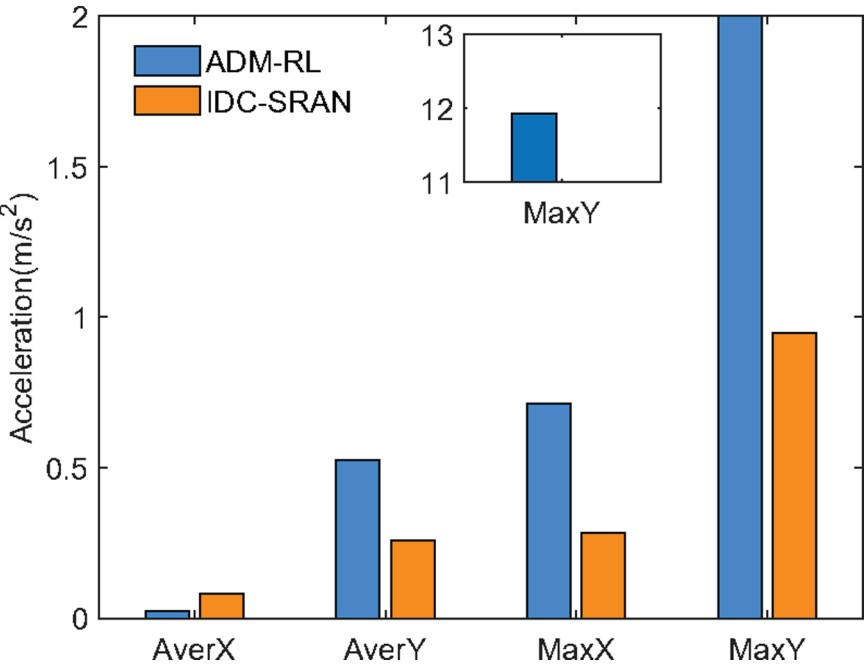

**Fig 13. The bar chart comparing the mean and maximum absolute acceleration of both IDC-SRAN and ADM-RL.**

the robot dynamics framework developed in this paper. Consequently, this part does not conduct further analysis on motor wear or energy efficiency concerns.

Compared to robots that remain stationary, those that adjust their velocity to avoid dynamic pedestrians exhibit behaviors more akin to natural human evasive habits. Fig 15 illustrates the relative distance variations between the robot and the four nearest pedestrians, with blue and red dashed lines demarcating the boundaries of comfortable and intimate distances as per human social norms, which are 1.2 m and 0.5 m, respectively. Due to the fact that the distance sequence of pedestrians relative to robots in the simulated environment constantly changes between two adjacent moments, Fig 15 presents a discrete point form. During the approximately 10–12-second interval, the nearest pedestrian progressively approached the robot. To avoid excessively compromising the pedestrian's comfort boundaries, the robot proactively reduced its lateral velocity while increasing longitudinal velocity. This adaptive control behavior aligns with the intelligence demonstrated in the earlier analysis of Fig 11(a). This phenomenon also confirms that the collision penalty (Eq 22) effectively guides the agent to make reasonable torque adjustments to avoid excessive penalties, which further demonstrates the intelligence of the IDC-SRAN framework.

Fig 16 presents detailed views of human-robot interaction trajectories from repeat experiments in Scene 1, where the black trajectory represents the robot's path. During critical time intervals, the robot conducts evasive maneuvers to ensure safe human-robot interaction.

This study focuses on enhancing the feasibility of robotic decision-control strategies through the deployment of dynamics models, while not addressing perception-related research. Consequently, practical deployment of the current IDC-SRAN framework still faces challenges. Considering that Sim2Real challenges for decision-control frameworks primarily stem from aspects such as real-time, feasibility and scene generalization, and the feasibility

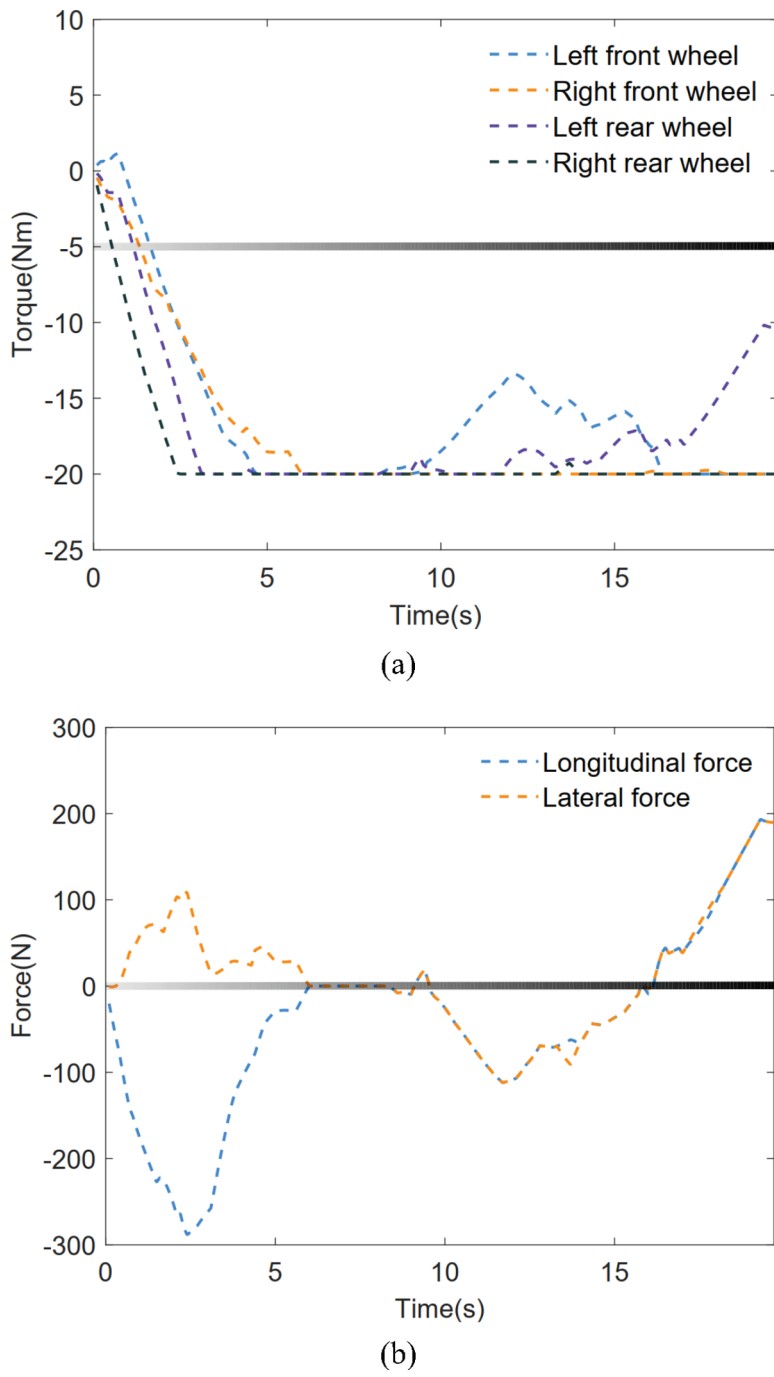

**Fig 14. Variation curve of robot driving force and four wheel torque under IDC-SRAN framework in Scene 1.**

has been proven. Therefore, this paper conducts multiple repeated simulation experiments in Scene 1 and 2, complemented by statistical analysis of computation time (including the combined computational time of environmental states and policy networks), to indirectly validate the framework's practical applicability. It's no doubt that this compromise validation approach

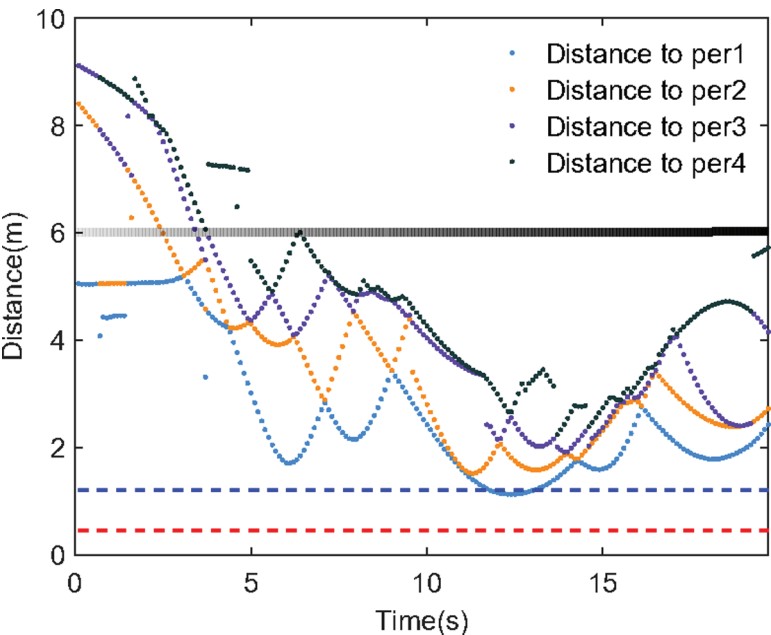

**Fig 15. The relative distance between the robot and the four nearest pedestrians.**

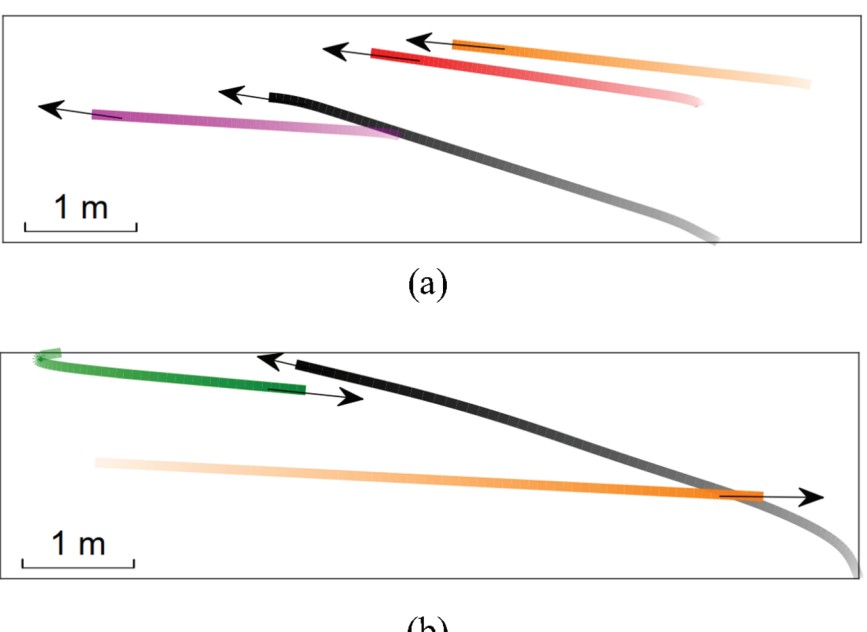

**Fig 16. The detailed views of human-robot interaction trajectories from repeat experiments in Scene 1.**

has limitations: it does not account for unknown disturbances in sensor perception and overlooks discrepancies between simulated conditions (robot models, pedestrian kinematics, environments) and real-world implementations. These gaps will be addressed in future research.

**Table 1. Collision rates and completion rates across different scene.**

| Scene | Method | Computing time | Completion rate | Collision rate | |
|---|---|---|---|---|---|
| | | | | Person | Wall |
| Scene 1(15) | ADM-RL | 0.009 s | 94 | 4 | 2 |
| | IDC-SRAN | 0.011 s | 91 | 7 | 2 |
| Scene 2(10) | ADM-RL | 0.009 s | 92 | 5 | 3 |
| | IDC-SRAN | 0.011 s | 90 | 6 | 4 |

Table 1 presents the pass rates and average computation times of different algorithms under repeated simulations across multiple scene. Specifically, Scene 1 involves 15 pedestrians with randomized initial and goal positions, while Scene 2 includes 10 such pedestrians. In all repeated simulation, pedestrian motion trajectories remain distinct. This methodology ensures the randomicity and statistical validity of simulation experiments through systematic randomization. It can be observed that compared to ADM-RL, the completion rate of IDC-SRAN incorporating the dynamic model exhibits a slight decrease. This stems from the fact that the motions of IDC-SRAN adhere to dynamic characteristics, eliminating abrupt action adjustments. Consequently, delayed collision avoidance maneuvers may occur, leading to an increased collision rate. However, when weighed against the substantial reduction in acceleration magnitudes (indicative of enhanced feasibility), this marginal loss in completion rate remains operationally acceptable. Simultaneously, the expanded state space (incorporating velocity and actual torques) results in moderately elevated computational demands. Nevertheless, the computational speed remains within practical limits, enabling real-time given the simulation step size of 0.4 seconds. Notably, both ADM-RL and IDC-SRAN demonstrate scenario generalization capabilities, as the completion rate of Scene 2 did not show a significant decrease.

## 5 Conclusion

This study proposes an integrated decision-control framework for social robot autonomous navigation (IDC-SRAN), which ensures the feasibility of the RL strategy by employing incremental wheel torques. Firstly, the CIIRL framework was proposed to explore the explicit and implicit reward mechanisms of pedestrian walking norms, which enhances the socially-aware of robots when performing tasks in complex social environments. In the Advantage Actor-Critic algorithm, the dynamics model of a four-mecanum-wheel Robot has been incorporated. The system actions are modified to incremental torques, and the actual torques along with the lateral and longitudinal velocities are incorporated into the state space. This direct association with the robot controller lays the foundation for an integrated decision-control strategy, thereby enhancing its feasibility. Taking into account the nonlinear characteristics of robots and integrating the dynamic characteristics of the whole vehicle, a driving-force-guided reward function has been designed to alleviate non-optimal behaviours caused by model transient transition processes. Finally, experiments were conducted in a simulation environment constructed on the Webots platform. The results demonstrated that IDC-SRAN achieves a peak acceleration of merely 8.3% of the baseline method while maintaining a multi-scenario task completion rate exceeding 90%. These findings validate the intelligence and feasibility of the proposed IDC-SRAN framework.

The future work of this study will focus on refining the model: simplified model assumptions often lead to accuracy degradation, which adversely impacts control performance. Given that enhancing data quality and quantity is critical for advancing the effectiveness of Inverse

Reinforcement Learning (IRL), this will remain a key priority in our future research. Additionally, this study will further investigate the previously overlooked visual module to enable practical deployment of the decision-control algorithms.

## Supporting information

**Expert demonstration. This excel contains all expert demonstration data.**
(XLSX)

## Author contributions

**Conceptualization:** Hui Li, Mingyue Luo.

**Data curation:** Mingyue Luo, Shuofeng Cong.

**Funding acquisition:** Wanbo Luo.

**Investigation:** Mingyue Luo.

**Methodology:** Hui Li, Mingyue Luo.

**Project administration:** Wanbo Luo.

**Software:** Mingyue Luo.

**Supervision:** Hewei Li.

**Validation:** Mingyue Luo, Hewei Li.

**Visualization:** Mingyue Luo.

**Writing – original draft:** Mingyue Luo.

**Writing – review & editing:** Hui Li, Wanbo Luo, Hewei Li, Shuofeng Cong.

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
