## [Decision Letter · Decision Letter 0]

PONE-D-25-07505Integrated Decision-Control for Social Robot Autonomous Navigation Considering Nonlinear Dynamics ModelPLOS ONE

Dear Dr. Luo, Thank you for submitting your manuscript to PLOS ONE. After careful consideration, we feel that it has merit but does not fully meet PLOS ONE’s publication criteria as it currently stands. Therefore, we invite you to submit a revised version of the manuscript that addresses the points raised during the review process.

Specifically, the introduction should include details of the research gap to be solved. The contributions of this work should be clearly clarified. More details about the experiments should be given.

We look forward to receiving your revised manuscript.

Kind regards,

Ying Shen, Ph.D.

Academic Editor

PLOS ONE

Journal Requirements:

“This work support from the Department of Science and Technology of Jilin Province (grant number: 20220204090YY).”

“The authors gratefully acknowledge support from the Department of Science and Technology of Jilin Province (grant number: 20220204090YY).”

“This work support from the Department of Science and Technology of Jilin Province (grant number: 20220204090YY).”

5. Please ensure that you refer to Figure 3 in your text as, if accepted, production will need this reference to link the reader to the figure.

Reviewers' comments:

Reviewer's Responses to Questions

**Comments to the Author**

1. Is the manuscript technically sound, and do the data support the conclusions?

Reviewer #1: Yes

Reviewer #2: Yes

Reviewer #3: Partly

Reviewer #4: Yes

2. Has the statistical analysis been performed appropriately and rigorously? 

Reviewer #1: Yes

Reviewer #2: Yes

Reviewer #3: No

Reviewer #4: I Don't Know

3. Have the authors made all data underlying the findings in their manuscript fully available?

Reviewer #1: Yes

Reviewer #2: Yes

Reviewer #3: No

Reviewer #4: Yes

4. Is the manuscript presented in an intelligible fashion and written in standard English?

Reviewer #1: Yes

Reviewer #2: Yes

Reviewer #3: Yes

Reviewer #4: Yes

5. Review Comments to the Author

Reviewer #1: 1: The contributions of the article are not completely clear. A more precise description of the contributions should be included.

2: The choice of the Four-Mechanism-Wheel Robot (FMWR) for the experiments is unjustified.

3: Line 159 is unclear; something is missing after the colon.

4: In equations 19 and 20, elements are not separated by the appropriate space, and the words used to express the cases are not understood.

5: The experiments carried out in simulation do not mention the limitations of the simulated environment.

6: Advantage Actor-Critic (A2C) is used for decision-making, but it is not justified why it is chosen over other reinforcement learning strategies, such as PPO or SAC, which are also popular in autonomous navigation problems.

Reviewer #2: This article presents a valuable contribution to social robot autonomous navigation through a novel Integrated Decision-Control Framework for Social Robot Autonomous Navigation (IDC-SRAN) that considers nonlinear dynamics models. The work addresses the important challenge of ensuring reinforcement learning strategy feasibility in real-world applications by bridging the dynamics mismatch between simulations and physical implementation. Although the methodology is well-conceived and demonstrates promising results, several aspects of the presentation, evaluation, and implementation details need significant improvement before publication. My comments are as follows:

1- The abstract effectively conveys the core contribution of the paper but contains several grammatical issues and redundancies. For example, "but lacks discussions on the feasibility of strategies" could be more precisely stated. The phrase "which takes into account the nonlinearity of the social robot model" appears twice in different forms. The technical approach is well summarized, but I recommend revising for clarity and conciseness. The statement about experiments "demonstrating intelligence and feasibility" is vague - specific metrics of improvement would strengthen this claim.

2- The introduction provides adequate background on RL for robot navigation but could benefit from a more focused discussion of the specific gap being addressed.

3- Lines 35-49: The discussion on nonlinear characteristics and transient behaviors is relevant but somewhat disconnected from the preceding paragraphs. Consider restructuring to create a more coherent narrative.

4- The contributions should be more explicitly stated, perhaps as bullet points, to help readers identify the novelty.

5- The literature review misses some key recent works on dynamics-aware RL for robotics and sim-to-real transfer, particularly those published in 2023-2025.

6- The introduction provides a good overview of reinforcement learning for robot navigation, but could benefit from incorporating insights from related domains where similar challenges have been addressed. The recent reviews like "Reinforcement learning in robotic applications: a comprehensive survey", "Review of deep reinforcement learning for robot manipulation", "Review of machine learning in robotic grasping control in space application" discusses challenges in bridging simulation-to-reality gaps in robotic control systems, which parallels your focus on dynamics mismatch issues. This reference would strengthen your discussion in lines 37-39 about the "significant gap from simulation to reality (sim2real)" and provide readers with additional context on how similar challenges manifest across different robotic applications.

7- The paper's hierarchical structure is unclear in several places, making it difficult to follow the logical flow of ideas. I recommend implementing a consistent numbering system (e.g., 1, 1.1, 1.1.1) or clear typographical distinctions between section levels to improve readability and help readers navigate the paper's structure.

8- The kinematic model in Equation (1) is a reasonable foundation, but the analysis would benefit from explaining the practical implications of these equations for real-world control. How do errors in this model affect controller performance?

9- The mass matrix M(q) in Equation (5) assumes uniform mass distribution, but real robots often have asymmetric mass distributions. This simplification should be acknowledged as a potential limitation.

10- The Coriolis and centrifugal force matrix C(q,q̇) in Equation (6) includes terms for platform rotation but doesn't address how wheel-surface interactions might affect these forces during high-speed maneuvers.

11- Equation (8) introduces the nominal mass term M0, but doesn't discuss how the uncertainty in mass estimation affects control stability, which is critical for reliable navigation.

12- The state space representation (lines 164-177) includes robot and pedestrian positions/velocities but doesn't explicitly include uncertainty measurements, which are crucial for robust decision-making in dynamic environments.

13- The reward function learning approach using IRL is interesting, but the explanation of how the GOED dataset was created lacks sufficient detail - particularly about how final pedestrian positions were determined to be "goal positions" when this information wasn't explicitly available in the ETH dataset.

14- The CIIRL framework mentioned on line 184 is not clearly defined before being used.

15- The relationship between the Maximum Entropy Deep Inverse Reinforcement Learning (MEDIRL) in line 223 and the previously mentioned IRL approach isn't clearly established.

16- The driving-force-guided reward function in Equation (18) is a novel contribution, but there's no ablation study showing its specific impact compared to alternative formulations.

17- The constraint of actions to [-1,1] range (line 181) seems arbitrary - the physical justification for this specific range should be explained.

18- The paper lacks details on the simulator parameters - particularly how the Webots environment was configured to represent realistic pedestrian behaviors.

19- The comparative analysis between ADM-RL and IDC-SRAN frameworks is qualitative rather than quantitative - no statistical measures of performance improvement are provided.

20- The policy and value network architecture description (lines 295-298) is too brief - more details on layer sizes, activation functions, and training hyperparameters are needed for reproducibility.

21- The training process description doesn't specify the number of training iterations, computational resources used, or convergence criteria.

22- Figures 5(a) and 5(b) show training losses, but don't include validation losses, making it difficult to assess potential overfitting.

23- The velocity profiles in Figures 6(b) and 7(b) show qualitative improvements in smoothness, but lack quantitative metrics such as jerk measurements or energy consumption comparisons.

24- The robot trajectories in Figures 6(a) and 7(a) demonstrate different behaviors, but there's no analysis of path optimality or safety margins.

25- The robot-pedestrian distance measurements in Figure 9(a) show the robot maintains distances above the "intimate distance" threshold, but there's no statistical analysis of how consistently these safety distances are maintained.

26- The torque variation curves in Figure 8(a) show how the four wheels are controlled, but don't analyze the physical implications of these control signals - such as energy efficiency or potential motor wear.

27- The driving force analysis in Figure 8(b) is insightful, but doesn't examine how these forces translate to acceleration profiles and their effects on passenger comfort if this system were deployed in practical applications.

Reviewer #3: The manuscript is well written overall, however, some issues need to be resolved. The comments are as follows:

1 The abstract is concise while informative presenting the background, methods, and a summary of research process, however, there is a lack of key concrete values obtained from the experiment.

2 Expand the literature review to provide a more comprehensive background on Autonomous Navigation for Social Robots in similar studies, including recent advancements.

3 Consider clearly stating the research objectives in the introduction to ensure the study's purpose is explicitly defined.

4 Provide figures of the constructed scenario and the experiment process.

5 Elaborate on why IDC-SRAN could ensure the intelligence and feasibility of social robot autonomous navigation. Including potential factors influencing the experimental outcomes would strengthen the discussion.

6 It is recommended to perform validation experiments on a real test platform to evaluate the autonomous navigation capabilities of the proposed Framework.

Reviewer #4: Suggestion for modification:

Further explore how to improve the effectiveness of IRL.

Design a more comprehensive reward function to better meet the needs of autonomous navigation for social robots.

Conduct experimental verification on actual robot platforms to validate the effectiveness of the IDC-SRAN framework in practical applications.

Improve the references in the paper and ensure that the citation format is correct.

Final decision:

Suggest hiring after modification.

6. PLOS authors have the option to publish the peer review history of their article (what does this mean?). If published, this will include your full peer review and any attached files.

Reviewer #1: No

Reviewer #2: No

Reviewer #3: No

Reviewer #4: No

---

## [Author Response · Author response to Decision Letter 1]

8 Apr 2025

Dear editors and reviewers,

On behalf of all the authors, I would like to extend our sincere gratitude for the time and effort you have dedicated to reviewing our manuscript. We fully understand that enhancing the quality of the manuscript is of utmost importance, and we highly appreciate the valuable contributions made by the reviewers. Your insightful comments and suggestions have been instrumental in guiding us towards refining our work.

We have meticulously addressed each of your concerns and have provided a detailed, point-by-point response to your feedback. In our revised manuscript, you will find that the changes and additions corresponding to your comments are highlighted in red for easy identification. This approach is intended to facilitate a clear and straightforward review process, allowing you to see precisely how we have incorporated your suggestions and improved the manuscript accordingly.

Once again, we thank you for your meticulous review and constructive input. We are confident that these revisions have significantly enhanced the clarity and robustness of our research. We look forward to your continued guidance and support as we strive to achieve the highest standards of academic excellence. For detailed responses, please refer to the attached document: Response to Reviewers.

Yours sincerely,

Mingyue Luo

E-mail: 1202101007@stu.ccut.edu.cn

---

## [Decision Letter · Decision Letter 1]

Integrated Decision-Control for Social Robot Autonomous Navigation Considering Nonlinear Dynamics Model

PONE-D-25-07505R1

Dear Mingyue Luo,

We’re pleased to inform you that your manuscript has been judged scientifically suitable for publication and will be formally accepted for publication once it meets all outstanding technical requirements.

Kind regards,

Ying Shen, Ph.D.

Academic Editor

PLOS ONE

Additional Editor Comments (optional):

Reviewers' comments:

Reviewer's Responses to Questions

**Comments to the Author**

1. If the authors have adequately addressed your comments raised in a previous round of review and you feel that this manuscript is now acceptable for publication, you may indicate that here to bypass the “Comments to the Author” section, enter your conflict of interest statement in the “Confidential to Editor” section, and submit your "Accept" recommendation.

Reviewer #2: All comments have been addressed

Reviewer #3: All comments have been addressed

Reviewer #4: All comments have been addressed

2. Is the manuscript technically sound, and do the data support the conclusions?

Reviewer #2: (No Response)

Reviewer #3: Yes

Reviewer #4: Yes

3. Has the statistical analysis been performed appropriately and rigorously? 

Reviewer #2: (No Response)

Reviewer #3: Yes

Reviewer #4: Yes

4. Have the authors made all data underlying the findings in their manuscript fully available?

Reviewer #2: (No Response)

Reviewer #3: Yes

Reviewer #4: Yes

5. Is the manuscript presented in an intelligible fashion and written in standard English?

Reviewer #2: (No Response)

Reviewer #3: Yes

Reviewer #4: Yes

6. Review Comments to the Author

Reviewer #2: (No Response)

Reviewer #3: The authors have addressed all the comments in the revised version of the paper. I would suggest the acceptance of the paper.

Reviewer #4: Nonlinear behavior model is considered: compared with the traditional simplified model of robot navigation, IDC - SRAN can better adapt to the real environment and improve the feasibility and robustness of navigation.

IRL-based pedestrian norm learning: IRL-based pedestrian norm learning enables robots to better understand and adapt to human behavior patterns and improve their performance in social environments.g.

Drive-Guide Reward Function: guides the robot to explore more appropriately during transient transitions by rewarding the drive in the same direction as the target, thereby improving mission efficiency and safety.g.

Experimental results: The experimental results show that the peak acceleration achieved by IDC-SRAN is about 8.3% of the baseline method, and the task completion rate exceeds 90%, which further verifies the intelligence and robustness of the framework.

Recommend further changes to the text language

7. PLOS authors have the option to publish the peer review history of their article (what does this mean?). If published, this will include your full peer review and any attached files.

Reviewer #2: No

Reviewer #3: No

Reviewer #4: No

---

## [Editor Report · Acceptance letter]

PONE-D-25-07505R1

PLOS ONE

Dear Dr. Luo,

I'm pleased to inform you that your manuscript has been deemed suitable for publication in PLOS ONE. Congratulations! Your manuscript is now being handed over to our production team.

Kind regards,

on behalf of

Dr. Ying Shen

Academic Editor

PLOS ONE